# The Vasoactive Role of Perivascular Adipose Tissue and the Sulfide Signaling Pathway in a Nonobese Model of Metabolic Syndrome

**DOI:** 10.3390/biom11010108

**Published:** 2021-01-15

**Authors:** Sona Cacanyiova, Samuel Golas, Anna Zemancikova, Miroslava Majzunova, Martina Cebova, Hana Malinska, Martina Hüttl, Irena Markova, Andrea Berenyiova

**Affiliations:** 1Center of Experimental Medicine, Institute of Normal and Pathological Physiology, Slovak Academy of Sciences, 841 04 Bratislava, Slovakia; samuel.golas@savba.sk (S.G.); anna.zemancikova@savba.sk (A.Z.); miroslava.majzunova@savba.sk (M.M.); martina.cebova@savba.sk (M.C.); andrea.berenyiova@savba.sk (A.B.); 2Department of Animal Physiology and Ethology, Faculty of Natural Sciences, Comenius University, Bratislava, 811 08 Bratislava, Slovakia; 3Center for Experimental Medicine, Institute for Clinical and Experimental Medicine, 140 21 Prague, Czech Republic; haml@ikem.cz (H.M.); mabw@ikem.cz (M.H.); irma@ikem.cz (I.M.)

**Keywords:** perivascular adipose tissue, H_2_S, isolated artery, Wistar, HTG, metabolic syndrome

## Abstract

The aim of this study was to evaluate the mutual relationship among perivascular adipose tissue (PVAT) and endogenous and exogenous H_2_S in vasoactive responses of isolated arteries from adult normotensive (Wistar) rats and hypertriglyceridemic (HTG) rats, which are a nonobese model of metabolic syndrome. In HTG rats, mild hypertension was associated with glucose intolerance, dyslipidemia, increased amount of retroperitoneal fat, increased arterial contractility, and endothelial dysfunction associated with arterial wall injury, which was accompanied by decreased nitric oxide (NO)-synthase activity, increased expression of H_2_S producing enzyme, and an altered oxidative state. In HTG, endogenous H_2_S participated in the inhibition of endothelium-dependent vasorelaxation regardless of PVAT presence; on the other hand, aortas with preserved PVAT revealed a stronger anticontractile effect mediated at least partially by H_2_S. Although we observed a higher vasorelaxation induced by exogenous H_2_S donor in HTG rats than in Wistar rats, intact PVAT subtilized this effect. We demonstrate that, in HTG rats, endogenous H_2_S could manifest a dual effect depending on the type of triggered signaling pathway. H_2_S within the arterial wall contributes to endothelial dysfunction. On the other hand, PVAT of HTG is endowed with compensatory vasoactive mechanisms, which include stronger anti-contractile action of H_2_S. Nevertheless, the possible negative impact of PVAT during hypertriglyceridemia on the activity of exogenous H_2_S donors needs to be taken into consideration.

## 1. Introduction

Perivascular adipose tissue (PVAT) represents a specific type of adipose tissue deposit surrounding blood vessels. In addition to providing mechanical protection, PVAT seems to be an important secretory organ because of its ability to release biologically active molecules. The results obtained in recent years have indicated that, under physiological conditions, PVAT exerts predominantly anticontractile effects that are induced by a transferable factor called adipocyte-derived relaxing factor (ADRF), among other factors. Schleifenbaum et al. [1] proposed that ADRF could be hydrogen sulfide (H_2_S), an important gaseous transmitter. They observed in mesenteric arteries and aortas of normotensive rats and mice that the inhibition of endogenous H_2_S had no effect on vessels without perivascular fat, whereas the inhibition of endogenous H_2_S significantly reversed the anticontractile effect of arteries with PVAT. Similarly, Fang et al. [2] found in rat aortas that blocking the K_ATP_ channel abolished the anticontractile effect of endogenous H_2_S released from PVAT. Nevertheless, in addition to PVAT [2], H_2_S can be produced by arterial smooth muscle cells [3] and endothelial cells [4], and it has been confirmed that H_2_S exerts both vasoconstrictor and vasorelaxant effects in rat conduit arteries [5]. Moreover, contradictory results regarding the synergistic and antagonistic effects of H_2_S and nitric oxide (NO) have been published. Coletta et al. [6] showed that NO and H_2_S are mutually required for the physiological control of vascular function. The authors confirmed that pretreatment with an H_2_S donor potentiated the vasorelaxant response of the thoracic aorta to acetylcholine and to an NO donor and increased cGMP levels. On the other hand, Kubo et al. [7] showed that the H_2_S donor induced the inhibition of endothelial NO-synthase (eNOS) activity in the arterial walls of rat and mouse aortas and that this effect was associated with an increase in arterial tension. Moreover, antioxidant effects of H_2_S that could be mediated through transcription factors such as Nrf2 and nuclear factor-κB were observed. H_2_S can stimulate enzymatic and nonenzymatic endogenous antioxidants through these transcription factors in different parts of the cardiovascular system [8]. Therefore, endogenous H_2_S can operate in vascular tone control and blood pressure regulation in distinct ways.

Both PVAT and the sulfide signaling pathway can interfere with the etiopathogenesis of different cardiovascular and metabolic diseases. Our previous findings showed that the PVAT–H_2_S interaction might play an important role in the modulation of vascular function in spontaneously hypertensive rats (SHRs) [9]. We confirmed the procontractile activity of H_2_S produced by the arterial wall as a probable pathologic feature of SHRs. Contemporarily, PVAT of the mesenteric artery was endowed with compensatory vasoactive mechanisms that included stronger anticontractile activity and potentiation of exogenous H_2_S vasorelaxation. Obesity and diabetes could be associated with adipose tissue inflammation and dysregulation of adipokine production, leading to the impaired anticontractile effect of PVAT [10]. However, the effect of obesity or metabolic state on the H_2_S system in adipose tissue is controversial. Beltowski [11] showed that, in obese rats fed a high-calorie diet for 3 months, the anticontractile effect of PVAT was impaired, and this impairment was associated with lower expression and activity of cystathionine-γ-lyase (CSE), an H_2_S-producing enzyme, as well as reduced H_2_S production by PVAT. Moreover, rats were characterized by hyperlipidemia and insulin resistance. In contrast, the anticontractile effect of PVAT was enhanced in obese animals fed a high-calorie diet for 1 month. These animals had a normal plasma lipid profile, insulin concentration, and insulin sensitivity, as well as unchanged CSE expression; however, H_2_S production by PVAT measured ex vivo was higher than that in control animals. From this point of view, two hypotheses arise: (a) a stimulated sulfide signaling pathway in PVAT acts as a part of compensatory vasoactive mechanisms and/or (b) the degree of metabolic disorder affects the vasoactive activity of PVAT and H_2_S. To test these two hypotheses, we used a nonobese model of genetically fixed hypertriglyceridemia: the hereditary hypertriglyceridemic (HTG) rat. This unique rat strain selected from Wistar rats exhibits most of the symptoms of metabolic syndrome and represents a suitable model for the study of dyslipidemia, insulin resistance, and prediabetes [12,13]. Disturbed redox balance has been observed in HTG rats, and this disturbance combined with the changed bioavailability of NO could contribute to the development of endothelial dysfunction [13,14]. However, although this model is also characterized by vascular complications and mild hypertension [15], the role of PVAT and the sulfide signaling pathway and their mutual interaction in vasoactive responses have not been investigated to date. The aim of this study was to evaluate and compare the participation of PVAT and endogenous H_2_S in the contractile and relaxant responses of arteries isolated from normotensive Wistar rats and HTG rats. We also evaluated the vasoactive effects of exogenous H_2_S, expression of CSE and eNOS, and total activity of NOS. The redox state in the cardiovascular system was determined through measurement of superoxide levels and the expression levels of antioxidant enzymes (superoxide dismutases, SODs) in the aortic wall and PVAT. Tissue injury of the arcus aortae in HTG rats was also evaluated.

## 2. Materials and Methods

### 2.1. Guide for the Use and Care of Laboratory Animals

Animals were bred in accordance with the institutional guidelines of the State Veterinary and Food Administration of the Slovak Republic and the Committee on the Ethics of Procedures in Animal, Clinical, and Other Biomedical Experiments (Permit Number: EC/CEM/2017/4) of the Centre of Experimental Medicine, as well as in accordance with the European Convention for the Protection of Vertebrate Animals used for Experimental and other Scientific Purposes, Directive 2010/63/EU of the European Parliament. The Wistar and HTG rats used in this study were received from the accredited breeding wholesale company Velaz, s.r.o. (Czech Republic) and the Institute of Clinical and Experimental Medicine (Czech Republic), respectively, and were housed under a 12 h light/12 h dark cycle at a constant humidity (45–65%) and temperature (20–22 °C) with free access to standard laboratory rat chow (Altromin, Germany; maintenance diet for rats and mice; pellets) and drinking water (ad libitum).

### 2.2. Experimental Animals and Basic Parameters

Eighteen- to 20-week-old male Wistar rats (n = 8) and HTG rats (n = 9) were used in this study. The body weight (BW) of each rat was determined before decapitation. The systolic blood pressure (SBP) was measured in prewarmed rats by noninvasive plethysmography of rat tail arteries before the beginning of the in vitro study. At the end of the study, animals were sacrificed by decapitation after brief anesthetization with CO_2_, the heart and retroperitoneal fat were weighed, and the tibia length was measured. The thoracic aorta (TA) and the superior mesenteric artery (MA) were isolated for further functional examination in vitro. After decapitation, aliquots of serum and tissue samples were rapidly removed, weighed, frozen in liquid nitrogen, and stored at −80 °C for biochemical analysis. Serum levels of glucose and lipids were measured using commercially available kits (Erba Lachema, Brno, Czech Republic; Roche Diagnostics, Mannheim, Germany). Serum MCP-1, TNFα, IL-6, leptin, and hs-CRP concentrations were determined using rat ELISA kits (MyBioSource, San Diego, CA, USA; eBioscience, San Diego, CA, USA; Alpha Diagnostics International, San Antonio, TX, USA). To measure triacylglycerols in tissues, samples were extracted in chloroform/methanol. Alanine-aminotransferase, creatinine, and urea were commercially determined in plasma in Laboklin GMBH (Bad Kissingen, Germany). The resulting pellet was dissolved in isopropyl alcohol, after which the triacylglycerol content was measured by enzymatic assay (Erba Lachema). For the oral glucose tolerance test (OGTT), blood glucose was determined after intragastric administration of a glucose load (300 mg/100 g b.wt.) following overnight fasting. Blood was drawn from the tail before the glucose load at 0 min and thereafter at 30, 60, and 120 min.

### 2.3. Functional Study

The vessels were divided into two groups, vessels without PVAT (PVAT−) and vessels with intact PVAT (PVAT+), to distinguish between the contribution of H_2_S produced by PVAT and the effect of total H_2_S produced by the arterial wall and surrounding perivascular fat. Subsequently, the TA (descending part of TA beginning below arcus aortae) and MA (superior MA beginning by the proximal part) were isolated and cleaned of connective tissue and cut into 5 mm (TA) and 2 mm (MA) length rings. In the case of PVAT−rings (without PVAT), under a microscope, the perivascular fat was removed from arterial surface with fine scissors, with caution not to damage the adventitia. In the case of PVAT+ rings (with PVAT), a continuous layer of perivascular fat (1 to 1.2 mm in width) was left around the vessel. The TA rings were vertically fixed between two stainless steel wire triangles, and the MA rings were mounted on thin wires and horizontally fixed. The TA and MA rings were immersed in a 20 mL and 10 mL incubation organ bath, respectively, containing Krebs solution (in mmol/L: 118 NaCl, 5 KCl, 25 NaHCO_3_, 1.2 MgSO_4_, 1.2 KH_2_PO_4_, 2.5 CaCl_2_, 11 glucose, 0.032 CaNa_2_EDTA). The solution was oxygenated with 95% O_2_ and 5% CO_2_ and kept at 37 °C. The upper wire triangles affixed to the TA ring were connected to isometric tension sensors (FSG-01, MDE, Budapest, Hungary), and changes in tension were registered by an NI USB-6221 AD converter (National Instruments, Austin, TX, USA and MDE, Budapest, Hungary). Changes in isometric tension were registered by S.P.E.L. Advanced Kymograph software (MDE, Budapest, Hungary). For the MA rings, the changes in isometric tension were measured using electromechanical transducers (MDE, Budapest, Hungary) and registered using an AD converter and DEWETRON software (DEWETRON, Prague, Czech Republic). A resting tension of 1 g was applied to each ring and maintained throughout a 45- to 60 min equilibration period until stress relaxation no longer occurred.

KCl (125 mmol/L, physiological Krebs solution changed to a solution in which NaCl was exchanged for equimolar concentration of KCl) was added to the organ bath for only 2 min to confirm the sufficient contractility of the sample. The presence of functional endothelium was assessed in all preparations by determining the ability of acetylcholine (Ach; 10^−5^ mol/L) to induce relaxation of noradrenaline (NA, 10^−6^ mol/L, 10^−7^ mol/L)-precontracted arteries. After washing with physiological Krebs solution and an equilibration period, experiments with NA started to detect the contractile responses. Adrenergic contractions were determined in TA as the responses to cumulatively applied exogenous NA (10^−10^–10^−5^ mol/L). The contractile responses were expressed as the active wall tension in g and normalized to the length of respective ring preparation (mm). The individual responses induced by increasing concentrations of NA have also been expressed as percentages of the maximal reached response induced by NA to demonstrate the sensitivity of the contractile apparatus. The adrenergic contractions of MA in response to endogenous NA were determined as the neurogenic responses elicited by electrical stimulation of periarterial sympathetic nerves. For transmural nerve stimulation (TNS), arterial rings were mounted between two platinum electrodes placed on either side of the preparation and connected to an ST-3 electrostimulator (MDE, Budapest, Hungary). Frequency–response curves to electrical stimuli were obtained using square pulses of 0.2 ms in duration at a supramaximal voltage of 35 V at 2–64 Hz for a period of 20 s. The TNS responses were pharmacologically tested in preliminary experiments. These contractions were abolished by tetrodotoxin, indicating the neurogenic nature of the responses. Guanethidine or phentolamine abolished the responses at all frequencies of stimulation. Therefore, we believe that the contractile responses were due to the release of NA from depolarized perivascular nerves. To examine the endothelium-dependent vasorelaxation, PVAT−/PVAT+ preparations of TA were first pre-contracted by concentration of NA, inducing a submaximal vasoconstriction, which was determined from the dose–response curve (10^−6^ mol/L). When the contraction reached a plateau, increasing concentrations of Ach were applied in a cumulative manner (10^−10^–10^−5^ mol/L). The rate of relaxation was expressed as a percentage of the NA-induced contraction.

To evaluate the participation of endogenous H_2_S in the vasoactive responses of TA, the contractile responses induced by increasing concentrations of NA and the relaxation responses induced by Ach were detected before and 20 min after acute administration of DL-propargylglycine (PPG) (10 mmol/L), an inhibitor of cystathionine-γ-lyase (CSE, an enzyme that produces H_2_S).

Na_2_S·9H_2_O was used to evaluate the vasoactive effect of exogenous H_2_S. Na_2_S dissociates in water solution to Na^+^ and S^2−^, which reacts with H^+^ to yield HS^−^ and H_2_S. We use the term Na_2_S to encompass the total mixture of H_2_S, HS^−^, and S^2−^. The stock solution of Na_2_S (100 mmol/L) was prepared by dissolving Na_2_S in ultrapure deionized water (≥18 MΩcm) (Millipore, Darmstadt, Germany) and placing it in a −80 °C freezer. On the day of the experiment, the stock solution (100 µL) was thawed and mixed with Krebs solution. The direct vasoactive effects of Na_2_S were observed on NA-precontracted (10^−7^ mol/L) TA rings by administration of increasing doses of Na_2_S (20, 40, 80, 100, 200, and 400 µmol/L). The rate of vasorelaxation is expressed as a percentage of the NA-induced contraction.

### 2.4. Total NO-Synthase Activity

Total NOS activity was determined in crude homogenates of the aorta and PVAT by measuring the formation of [^3^H]-l-citrulline from [^3^H]-l-arginine (ARC, Helena, MT, USA), as previously described and slightly modified by Pechanova et al. [16]. [^3^H]-l-citrulline was measured with the Quanta Smart TriCarb Liquid Scintillation Analyzer (Packard Instrument Company, Meriden, CT, USA).

### 2.5. Superoxide Levels

The level of superoxide anions was measured by a chemiluminescent method using lucigenin based on the intensity of the emitted photons. Heart and vessel samples were placed in Krebs solution immediately after removal. Oxygenated (mixed 95% O_2_, 5% CO_2_) 50 μmol/L lucigenin and samples in oxygenated Krebs solution were incubated for 20 min at 37 °C. After incubation, tissue samples were placed in scintillation vials with 50 μM lucigenin solution. Subsequent measurements were performed on a Tri-Carb 2910 TR Scintillation Detector (Perkin Elmer, Waltham, MA, USA). The resulting mean values were expressed as cpm/mg (“count per minute/milligram”) of tissue.

### 2.6. Western Blotting

PVAT and aortic samples were homogenized on ice in 0.05 M Tris-HCl buffer (pH 7.4) supplemented with protease inhibitors. The protein concentrations were determined with a BCA assay kit (MilliporeSigma, 71285). Proteins (20 µg total protein) were separated by 8%, 10%, or 15% SDS-PAGE depending on the size of the protein being measured and transferred to nitrocellulose membranes. The membranes were blocked with 5% milk in Tris-buffered saline containing Tween 20 (TBS-T). Afterwards, the membranes were incubated with a rabbit polyclonal anti-eNOS antibody (Abcam, Cambridge, UK; dilution 1:1000) overnight at 4 °C and with a mouse monoclonal anti-CSE antibody (Proteintech^®^, Manchester, UK; dilution 1:2000), rabbit polyclonal anti-SOD1 antibody (Santa Cruz Biotechnology, Dallas, TX, USA; dilution 1:2000), or rabbit polyclonal anti-SOD2 antibody (Abcam, Cambridge, UK; dilution 1:2000) for 2 h at room temperature. Following three washes (3 × 10 min) with TBS-T, the membranes were incubated with an anti-mouse IgG (H + L) peroxidase-conjugated antibody (Thermo Fisher Scientific, MA, USA; dilution 1:2000) and an anti-rabbit IgG HRP-linked antibody (Cell Signaling Technology, MA, USA; dilution 1:2000) for 1 h at room temperature. Both primary and secondary antibodies were diluted in TBS-T containing 1% milk. All blots were reprobed with a mouse monoclonal anti-GAPDH antibody (Santa Cruz Biotechnology, Dallas, TX, USA; dilution 1:2000) for 1 h at room temperature for PVAT samples or with a mouse monoclonal anti-α-actin antibody (Sigma-Aldrich, Saint Louis, MO, USA; dilution 1:1000) and incubated overnight at 4 °C for aorta samples. The signals were visualized with Clarity Western ECL Substrate (Bio-Rad, 1705061) using a ChemiDocTM Touch Imaging System (Bio-Rad) and quantified with Image Lab Software. Target protein amounts were normalized to GAPDH for PVAT samples or to α-actin for aorta samples and are presented in arbitrary units (a.u.).

### 2.7. Morphological Study

To evaluate an arterial tissue injury, the arcus aortae of HTG rats was excised and fixed with solution of 300 mmol/L glutaraldehyde in 100 mmol/L phosphate buffer post-fixed with 40 mM OsO_4_ in 100 mmol/L phosphate buffer, stained on block with 1% uranylacetate. After postfixation, the samples were dehydrated in an ascending series of alcohols and propylene oxide. The specimens were then embedded in Durcupan ACM. Thin sections of approximately 70 nm thickness, which were cut perpendicular to the longitudinal axis of artery on an ultramicrotome (Reichert Nova), were stained with lead citrate and examined in a Tesla BS 500 electron microscope.

### 2.8. Statistical Analysis

The data are expressed as the mean ± S.E.M. For the statistical evaluation of vasoactive responses between groups, three-way analysis of variance (ANOVA) with the Bonferroni post hoc test was used. To evaluate general cardiovascular, serum, and plasmatic parameters, including NO-synthase activity; superoxide levels; and eNOS, CSE, SOD1, and SOD2 expression, one-way ANOVA was used with the Bonferroni post hoc test. Differences between means were considered significant at *p* < 0.05.

### 2.9. Drugs

The following drugs were used: acetylcholine, sodium sulfide nonahydrate, and propargylglycine from Sigma-Aldrich (St Louis, MO, USA) and noradrenaline from Zentiva (Prague, Czech Republic). Acetylcholine, DL-propargylglycine, and noradrenaline were dissolved in distilled water, and sodium sulfide nonahydrate was dissolved in ultrapure deionized water.

## 3. Results

### 3.1. General Characteristics of Experimental Animals

Although the values of systolic blood pressure (SBP) were significantly increased in HTG rats compared with control Wistar rats, the body weight was lower and the heart weight and tibia length were unchanged in the HTG strain. The ratios of heart weight to body weight and heart weight to tibia length were significantly decreased, confirming a hypotrophy of the myocardium in the HTG rats. The level of TAG tended to be increased in the heart and kidney and was significantly increased in the serum of the HTG rats. We also confirmed impaired glucose tolerance, increased levels of leptin and nonesterified fatty acids, and decreased levels of HDL-C in the serum of the HTG rats. The levels of TNFα, IL-6, and hsCRP tended to be increased, and the MCP-1 level was significantly increased in the serum of the HTG rats. The levels of alanine-aminotransferase, creatinine, and urea in plasma were increased in the HTG rats compared with the Wistar rats (Table 1).

### 3.2. Functional Study

The application of Ach (10^−10^–10^−5^ mol/L) relaxed the NA-precontracted TA rings. Three-way ANOVA revealed a significant effect of PVAT (F_(1,268)_ = 113.28; *p* = 0) and strain (F_(1,268)_ = 122.03; *p* = 0) on the vasorelaxant response to Ach. PVAT significantly inhibited the vasorelaxant response in Wistar rats (*p* < 0.001) and HTG (*p* < 0.001) rats. Additionally, regardless of the presence of PVAT, Ach-induced relaxation was significantly (*p* < 0.001) reduced in the HTG rats (n = 9) compared with the Wistar rats (n = 8) (Figure 1a). In the Wistar rats, incubation with PPG had no effect on Ach-induced vasorelaxation in rings without PVAT or in rings with preserved PVAT (n = 8) (Figure 1b). On the other hand, PPG pretreatment significantly enhanced vasorelaxation in the HTG rats not only in PVAT− aortic rings (*p* < 0.001), but also in PVAT+ (*p* < 0.001) aortic rings (Figure 1c). In this strain, both the presence of PVAT (F_(1,344)_ = 302.52; *p* = 0) and incubation with PPG (F_(1,344)_ = 108.98; *p* = 0) had a significant effect on endothelium-dependent vasorelaxation.

Cumulative application of exogenous NA (10^−10^–10^−5^ mol/L) induced vasoconstriction in a concentration-dependent manner (n = 8). The presence of PVAT (F_(1,344)_ = 108.69; *p* = 0) and the strain (F_(1,305)_ = 358.75; *p* = 0) significantly affected these adrenergic vasocontractions. The HTG rats revealed a significantly increased contractile response to exogenous NA in both PVAT− (*p* < 0.001) and PVAT+ (*p* < 0.001) rings compared with the normotensive Wistar rats (Figure 2a). PVAT reduced the contractile responses in the Wistar rats (*p* < 0.01) and the HTG rats (*p* < 0.001). In the Wistar rats, PVAT had no effect on sensitivity to exogenous NA. On the other hand, not only PVAT− (*p* < 0.001), but also PVAT+ (*p* < 0.05) rings of HTG rats showed significantly increased sensitivity to NA compared with those of Wistar rats. The presence of PVAT reduced this parameter in the HTG rats (*p* < 0.001) (Figure 2b). Statistical analyses indicated a significant effect of PVAT (F_(1,301)_ = 16.58; *p* = 6.20 × 10^−5^) and strain (F_(1,301)_ = 69.97; *p* = 3.77 × 10^−15^) on sensitivity to NA. Acute pretreatment with PPG had no impact on adrenergic contraction in the Wistar rats (Figure 2c); on the other hand, it increased the maximal force of contractile responses in rings with preserved PVAT in the HTG rats (*p* = 0.01) (Figure 2d).

Transmural nerve stimulation (TNS, 2–64 Hz) induced the release of endogenous NA from nerve endings in the arterial wall of MA tissue, inducing contraction of smooth muscle cells. Generally, significant effects of strain (F_(1,147)_ = 5.61; *p* = 0.019) and PVAT (F_(1,147)_ = 40.1, *p* = 3.96 × 10^−9^) were shown in endogenous NA-induced contractions. In the Wistar rats, preserved PVAT had no effect on contractile responses after TNS. In the HTG rats, PVAT significantly increased contractions (*p* < 0.001); PVAT− rings from the HTG rats revealed a significantly reduced contraction compared with both PVAT− (*p* < 0.001) and PVAT+ (*p* < 0.001) mesenteric rings from the Wistar rats. However, the achieved contractions of PVAT+ rings from the HTG rats were comparable to the contractile responses recorded from PVAT− and PVAT+ rings from the Wistar rats (Figure 3).

The cumulative application of the H_2_S donor Na_2_S·9H_2_O (20, 40, 80, 10, 200, and 400 μmol/L) to NA-precontracted TA rings induced a dual effect in both Wistar rats and HTG rats. Concentrations of 20, 40, and 80 μmol/L induced contraction of the arterial wall, whereas concentrations of 100, 200, and 400 μmol/L evoked vasorelaxation of the arterial wall (Figure 4). In the Wistar rats, the presence of PVAT did not affect the dual vasoactive effect of the H_2_S donor. In the HTG rats, regardless of the presence of PVAT, we confirmed an increased maximal vasorelaxant phase of the Na_2_S-induced response (*p* < 0.01) compared with that in the Wistar rats. However, we recorded a significant effect of PVAT in HTG rats (*p* < 0.01) in connection with the contractile phase of the response; that is, a drift in favor of vasoconstriction was observed in aortic rings with preserved PVAT from HTG rats.

### 3.3. Protein Expression of eNOS and CSE and Total Activity of NOS

Differences in the protein expression of eNOS were observed in PVAT, but not in the aorta (Figure 5a,b). The expression of eNOS was significantly increased in the PVAT of the HTG rats (*p* < 0.01). Immunoblots detected a significant increase in CSE expression in the aorta (*p* < 0.05) and PVAT (*p* < 0.05) of the HTG rats compared with the control Wistar rats (Figure 5c,d).The total NOS activity was significantly decreased in both the aorta and PVAT of HTG rats by 60% (*p* < 0.01) and 46% (*p* < 0.05) of the control values, respectively (Figure 6).

### 3.4. Redox State in the Vascular System

Measurement of superoxide anions was used to evaluate oxidative stress in the cardiovascular system of HTG rats. Our results showed a significantly increased level of superoxide in the aorta of the HTG rats compared with the control Wistar rats (*p* < 0.05; Figure 7). The protein expression of SOD isoforms was used to determine the antioxidant response in the aorta and PVAT. Immunoblots showed significantly reduced protein expression of mitochondrial SOD2 (*p* < 0.05) without changes in SOD1 protein expression in the aorta (Figure 8a,c). Different results were observed in PVAT, in which the protein expression of cytosolic SOD1 was significantly stimulated (*p* < 0.01) and SOD2 expression did not show a significant difference (Figure 8b,d).

### 3.5. Morphological Study

We found a pronounced accumulation of products of lipid metabolism, which was localized mainly in the luminal part of the arterial wall (endothelial cells and subendothelial space). In the majority of endothelial cells, big electron-lucent lipid droplets were present and occupied the main part of the cytoplasm. Subendothelial space was enlarged and contained lipid droplets and abundant amorphous material (Appendix A). Less lipids were found in smooth muscle cells, and there was only a sporadic occurrence in intercellular space among smooth muscle cells (Appendix A).

## 4. Discussion

In our study, we demonstrated that, in HTG rats, PVAT and endogenous H_2_S could manifest a dual effect depending on the type of triggered signaling pathway. Sympathetic nerve stimulation led to the procontractile action of PVAT and H_2_S within the arterial wall contributed to endothelial dysfunction. On the other hand, PVAT of HTG is endowed with compensatory vasoactive mechanisms, which included stronger anti-contractile action of H_2_S. Finally, although H_2_S donors could represent prospective pharmacological tool, our results demonstrated that PVAT subtilized the vasorelaxant effect of H_2_S donor.

In our study, HTG rats were characterized by dyslipidemia, leading to ectopic lipid accumulation, impaired glucose tolerance, and insulin resistance of peripheral tissue. This rat strain also exhibits low-grade chronic inflammation that is typically present in states such as metabolic syndrome and prediabetes. Although the HTG rats were nonobese, we confirmed an increase in the amount of retroperitoneal fat. Moreover, we also observed mild hypertension accompanied by cardiac hypotrophy. Cardiac hypotrophy could be related to the fact that hypertriglyceridemia and metabolic syndrome are associated with increased lipid accumulation in nonadipose tissues. Ectopically stored lipids and their metabolites (diacylglycerols, ceramides, and fatty acyl-CoAs) in the liver, heart, muscles, and kidney lead to lipotoxicity, which may result in tissue injury [17,18]. We found increased levels of kidney and liver injury markers, such as alanine-aminotransferase, which also demonstrates myocardial damage [19]. In addition, our results confirmed endothelial dysfunction and increased sympathetic and adrenergic contractility as pathologic features of the cardiovascular system of HTG rats. Moreover, we revealed that, under conditions of hypertriglyceridemia, both PVAT and sulfide signaling pathways contributed to endothelial dysfunction; on the other hand, compensatory vascular mechanisms associated with the stronger anticontractile activity of H_2_S released from PVAT could also be triggered.

### 4.1. Procontractile vs. Anticontractile Actions of PVAT

In our study, we confirmed that the thoracic aortas of HTG rats, independent of the presence of PVAT, revealed increased adrenergic contractility compared with those of control rats. This finding is in agreement with our previous results, in which we found augmented vasoconstriction of the iliac artery in HTG rats associated with arterial wall thickening [15]. Nevertheless, as dyslipidemia could lead to impaired muscle metabolism and subsequent muscle damage [20], hypertrophy of smooth muscle cells within the arterial wall seems to be an unlikely explanation of increased contractility. In this study, we confirmed the increased sensitivity to exogenous noradrenaline in both PVAT−intact and PVAT−denuded arteries of HTG rats compared with those of control rats. High levels of triacylglycerols can modulate the cytosolic free Ca^2+^ concentration ([Ca^2+^]i) by affecting various aspects of Ca^2+^ handling in addition to enhancing Ca^2+^ influx and mobilization of Ca^2+^ stores [21]. Therefore, the increase in [Ca^2+^]i could very likely be responsible for enhanced vasoconstriction. On the other hand, despite the development of dyslipidemia, we surprisingly demonstrated a significant anticontractile activity of PVAT in the HTG rats that was higher than that in the Wistar rats. We confirmed a significant effect of the strain-PVAT interaction. HTG thoracic aortas with intact PVAT responded with lower intensity than PVAT−denuded HTG aortas and tended to decrease a contractile response to the level of response of normotensive rats with intact PVAT. Our observation is in contrast to data reported in the literature, which mostly showed a reduced anticontractile effect of PVAT in experimental models of metabolic disorder or diabetes [11,22]. However, the nonobese HTG rats used in this study were fed a standard diet and represent an early stage of metabolic syndrome and prediabetes in which metabolic changes have not reached a high degree of disorder, such as in the case of HTG rats treated with a high-fat diet [23]. We suggest that PVAT from nonobese HTG rats is able to develop compensatory vasoactive mechanisms against increased vascular tone that include the release of anticontractile factors. Similar findings were confirmed in spontaneously hypertensive rats (SHRs), in which PVAT of mesenteric arteries revealed a stronger anticontractile activity than those from normotensive rats to counteract increased vascular tone [9].

Moreover, we recorded the difference in the effect of PVAT when an endogenous source of noradrenaline was tested. Unlike exogenous noradrenaline, we demonstrated that transmural nerve stimulation led to procontractile properties of PVAT in mesenteric arteries from HTG rats. A detailed evaluation of the frequency–response curves again revealed a significant effect of strain and PVAT interaction. Török et al. [24] showed in adult normotensive rats that most of the sympathetic nerve terminals are concentrated in the surface layers of the proper mesenteric arterial wall. In our study, we documented that the procontractile effect of PVAT from HTG rats arose during neurogenic contractions, suggesting that active innervation could also be present in PVAT. Moreover, nerve stimulation could be associated with the overproduction of procontractile factors and oxidative stress. Lu et al. [25] and Gao et al. [26] showed in mesenteric arteries that PVAT promoted vasoconstriction to perivascular neuronal activation through the generation of angiotensin II and superoxide, and Knapp and Klann [27] confirmed that the presence of superoxide could induce long-lasting potentiation of synaptic transmission. Although the HTG rats in our experiment were not obese, we confirmed the increased amount of retroperitoneal fat together with the increased level of triacylglycerols in the plasma of the HTG rats. Hypertriglyceridemia alters the normal pattern in the population of lipoprotein subclasses, and as the level of plasma lipids increases, this may provide an increased quantity of substrate for lipid peroxidation [28]. Moreover, increased accumulation of lipids in adipocytes is associated with hypoxia and increased expression of proinflammatory cytokines, leading to macrophage infiltration and reactive oxygen species production [29,30]. In our study, we demonstrated increased superoxide formation and reduced protein expression of mitochondrial SOD, which suggests disturbed redox balance in arterial tissue of HTG rats compared with that of normotensive rats. In PVAT, we observed increased protein expression of cytosolic SOD, which can be elevated because of the presence of oxidative stress in this model [31], and can serve as a compensatory mechanism to maintain redox balance in the vasculature. Nevertheless, we suggest that the increased density of sympathetic innervation in the arterial wall and PVAT together with the increased oxidative stress, which were both generally demonstrated in HTG rats, could lead to a stronger procontractile effect of PVAT in HTG rats. Taken together, these data indicate that PVAT of HTG rats could manifest either procontractile or anticontractile properties, which could be tissue-specific. However, it is possible that both processes are mutually associated, and the pathologic effect of nerve stimulation could be counterbalanced by the stimulation of anticontractile properties.

### 4.2. Endothelial Function and PVAT

In the next part of the study, we evaluated acetylcholine-induced vasorelaxation as a marker of endothelial function. We confirmed that PVAT−intact arteries of both strains exerted a decreased vasorelaxant response, whereas the most inhibited vasorelaxation was in HTG rat arteries with intact PVAT. Our results are in agreement with the observation that hypertriglyceridemia in rats was associated with impaired endothelial function, which was accompanied by marked changes in vascular architecture [32,33]. Similar impairment of endothelium-dependent relaxation was also observed in mesenteric and carotid arteries, as well as in the mesenteric resistance arteries in Wistar rats given a diet that led to a selective elevation of triglycerides [34,35,36]. It seems that hypertriglyceridemia leads to endothelial dysfunction, and several mechanisms could be involved in this process, including the inhibition of the NO signaling pathway. In our study, we confirmed significantly inhibited NO-synthase (NOS) activity in both the arterial wall and PVAT, although the expression of the endothelial NOS (eNOS) isoform was significantly increased in PVAT. Pechanova et al. [37] also observed inhibition of NOS activity associated with increased expression of eNOS. Limited availability of NO has been shown to exert a negative feedback influence on eNOS expression via activation of nuclear factor κB, which could be responsible for increased transcription of eNOS. On the other hand, Kusterer et al. [35] showed that selective hypertriglyceridemia induced a progressive depression of endothelial vasodilator responsiveness that was associated not with a change in the expression of endothelial NOS, but rather with a marked increase in vascular superoxide anion production. Small dense particles, which are the main carriers of triglycerides [38], may themselves be prone to oxidation and could be responsible for impaired relaxation. Moreover, the evaluation of ultrastructural changes in the HTG aortic wall demonstrated the accumulation of products of lipid metabolism in the arterial wall, pronounced storages of lipids were localized mainly in luminal part of the arterial wall, and less lipids were found in smooth muscle cells, so there was a gradient in lipid deposition in the arterial wall. It seems that a relatively compact layer of internal elastic lamina acted as a protective coat to prevent the penetration of remnants of lipid metabolism deep into the tunica media. Similar results were described Kristek et al. [32], who also demonstrated accumulated products of lipolysis in endothelial cells and in the subendothelial space. All mentioned abnormalities might be important contributors to the endothelial dysfunction observed in HTG rats. Moreover, intact PVAT significantly worsened endothelium-dependent vasorelaxation in both strains. Although Payne et al. [39] demonstrated in coronary arteries of normotensive dogs that periadventitial adipose-derived factors can impair endothelial NO production via site-specific inhibition of NO-synthase phosphorylation, additional studies are needed to link the effects of PVAT−derived factors and endothelial function in normotensive conditions. Regarding the effect of PVAT under hypertriglyceridemic conditions, similar to our findings, Ketonen et al. [40] showed that, in the presence of PVAT, aortas of obese mice displayed impaired endothelium-dependent vasodilation that was restored after the removal of PVAT and after reducing superoxide and hydrogen peroxide formation. The authors confirmed that PVAT promoted endothelial dysfunction via mechanisms that were linked to increased NADPH oxidase-derived oxidative stress and increased production of proinflammatory cytokines. Dysfunctional PVAT induces vascular smooth muscle cells proliferation and endothelial dysfunction, which can be mediated by proinflammatory adipocytokines released by activated inflammatory cells in PVAT. Thus, the elevated circulating levels of MCP-1 in HTG rats can contribute to the development of endothelial dysfunction in the early state. Endothelial cells stimulated with MCP-1 activate adhesion molecules, leading to the proliferation and migration of leukocytes, which further promote the inflammatory process and the secretion of other proinflammatory cytokines, e.g., TNF❬ and IL-6. TNF❬ can decrease eNOS expression and increase the expression of endothelial adhesion molecules, and NFkB enhances ROS production by endothelial NADPH oxidases [41]. We suggest that pathological changes in adipose tissue, including an altered oxidative state and inflammation, could be responsible for the increased endothelial dysfunction confirmed in the HTG rats.

### 4.3. PVAT–H_2_S Interaction

The next question in this study was whether the degree of metabolic disorder affects the vasoactive effect and mutual relationship of PVAT and H_2_S. Fang et al. [2] confirmed CSE protein expression directly in adipocytes of PVAT. Using the method of methyl blue assay, the authors also showed that H_2_S, which was released from PVAT at an amount similar to that from aorta, acted as a vasodilator. In our previous study, we confirmed that, in SHRs, in which arterial hypertension was not associated with changes in triacylglycerol levels, PVAT of mesenteric arteries revealed a stronger anticontractile effect; however, this effect was mediated by an unknown factor other than H_2_S [9]. In this study, the inhibition of endogenous H_2_S reversed the anticontractile effect of PVAT in the thoracic aortas of HTG rats, indicating that the PVAT−released anticontractile factor (or one of them) could very likely be H_2_S. The highest expression of CSE, an H_2_S-producing enzyme, was observed in PVAT surrounding aortas in HTG rats that supported functional vasoactive responses. Similar results were shown by Beltowski and Wisniewska [10], who found that rats fed a high-calorie diet for 1 month, exhibited an enhanced anticontractile effect of PVAT and that H_2_S production by PVAT measured ex vivo was higher than that in control animals. Moreover, the authors also confirmed that short-term (1 month), but not long-term (3 month) feeding of rats with a high-fat diet led to the impairment of H_2_S oxidation and degradation as well as the augmentation of the H_2_S-mediated anticontractile effect of PVAT. Brancaleone et al. [42] demonstrated in a nonobese diabetic mouse model that, although the plasma level of H_2_S decreased as glycemia increased, the expression of CSE in arterial tissue as well as the vasorelaxant response to exogenous H_2_S were increased. Similar results were confirmed in this study; we observed an increase in CSE expression, as shown in the vasorelaxant effect of exogenous H_2_S. However, vasorelaxation in response to H_2_S was increased regardless of the presence of PVAT; moreover, the intact PVAT of aortas in HTG rats increased the contractile component of the vasomotor response to exogenous H_2_S. On the other hand, our previous results observed in SHRs confirmed a close relationship between the vasoactive effect of exogenous H_2_S and PVAT in essential hypertension [9]. In SHRs, we showed a reduced contractile component and an increased vasorelaxant component of responses induced by exogenous H_2_S in PVAT−intact arteries only. Given these results, we could conclude that, in HTG rats, which represent a nonobese model of metabolic syndrome and prediabetes, PVAT has the ability to trigger vasoactive compensatory mechanisms that include enhanced anticontractile activity associated with H_2_S action. On the other hand, although exogenous H_2_S donors have already been suggested as potential new antihypertensive therapeutics [43], the possible negative impact of hypertriglyceridemia and PVAT on the activity of exogenous H_2_S donors needs to be taken into consideration.

The last part of our study was focused on the role of endogenous H_2_S in the regulation of endothelial function. Our results showed that endogenous H_2_S participated in the inhibition of endothelium-dependent vasorelaxation in both PVAT−intact and PVAT−denuded rings in HTG rats. This result showed that, according to the type of reaction (vasoconstriction or vasorelaxation), the vasoactive activity of endogenous H_2_S could switch from anticontractile to antirelaxant. Similar results were found by Emilova et al. [22], who confirmed a dual precontractile/anticontractile vasoactive effect of endogenous H_2_S in hyperglycemic streptozotocin-induced diabetic rats. Whereas H_2_S exhibited an anticontractile effect at low concentrations of vasoconstrictor (serotonin), the effect of PVAT−released H_2_S at the highest concentrations of serotonin became vasoconstrictive and procontractile. In our study, we confirmed that endothelial dysfunction in HTG rats was associated with decreased NO-synthase activity, increased superoxide production, and a disturbed antioxidant response, which could reduce NO availability. On the other hand, Torok et al. [44] showed that NO production in HTG rats was further depressed by artificial inhibition of NOS with L-NAME, an inhibitor of NOS, demonstrating that the NO signaling pathway might not be fully inhibited. Our previous experiments using isolated thoracic aortas demonstrated in Wistar rats and SHRs that pretreatment with L-NAME diminished the contractile vasoactive effects of low concentrations of H_2_S [45,46]. Recently, Szijártó et al. [47] demonstrated that one of the major functions of H_2_S produced by CSE is to reduce endothelial NO bioavailability by the direct interaction of H_2_S and NO. Similarly, Kubo et al. [7] showed that H_2_S induced the inhibition of NOS activity in the arterial walls of rat and mouse aortas, and Geng et al. [48] showed that low doses of H_2_S donor downregulated the l-arginine/NO pathway by inhibiting eNOS expression and by decreasing NOS activity. Based on the findings, we hypothesize that H_2_S produced by the arterial wall and PVAT of HTG rats could inhibit residual NO signaling, leading to the attenuation of vasorelaxation. Therefore, the interaction between the sulfide and nitroso signaling pathways could likely participate in the endothelial dysfunction observed in HTG rats. On the other hand, the above documented anticontractile effect of H_2_S produced by PVAT could be explained by the activation of other signaling pathways, e.g., the stimulation of K_ATP_ or KCNQ channels in smooth muscle cells, leading to hyperpolarization and a subsequent decrease in [Ca^2+^]i [1,2]. We suggest that the effect of endogenous H_2_S could depend on activation of the appropriate signaling pathway and on the type of regulated vasoactive response. We cannot exclude that the vasoactive effects of endogenous H_2_S and PVAT could also be mediated through possible interaction with other regulatory systems (nerve, immune, and so on) and the mechanisms require further investigation.

## 5. Conclusions

In summary, our findings suggest that the PVAT–H_2_S interaction may play an important role in the modulation of vascular function in HTG rats. Both PVAT and endogenous H_2_S exhibited procontractile or anticontractile properties in isolated arteries depending on the type of signal pathway that was triggered. In the HTG rat, a nonobese model of metabolic syndrome, the procontractile effect of PVAT in the mesenteric artery was closely associated with perivascular nerve stimulation, and endogenousl H_2_S participated in endothelial dysfunction. On the other hand, PVAT of the thoracic aorta is probably endowed with compensatory vasoactive mechanisms including stronger anticontractile activity of H_2_S. Moreover, treatment with an exogenous H_2_S donor revealed significantly higher vasorelaxant effects in HTG thoracic aortas; however, the possible negative impact of hypertriglyceridemia (i.e., the increased contractile component of the response in PVAT−intact rings) also needs to be taken into consideration.

## Figures and Tables

**Figure 1 biomolecules-11-00108-f001:**
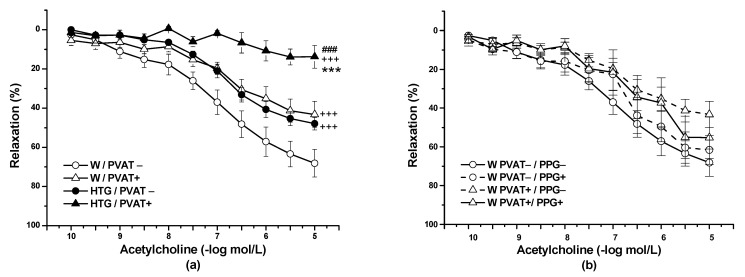
Endothelium-dependent vasorelaxation of the thoracic aorta of normotensive Wistar rats (W, n = 8) and hypertriglyceridemic rats (HTG, n = 9) with intact (PVAT+) or removed (PVAT−) perivascular adipose tissue (**a**) before (PPG−) and after (PPG+) pretreatment with propargylglycine (**b**,**c**). The results are presented as the mean ± S.E. M, and differences between groups were analyzed by three-way analysis of variance (ANOVA) with the Bonferroni post hoc test on ranks. *** *p* < 0.001 vs. W/PVAT+; +++ *p* < 0.001 vs. W/PVAT−; ### *p* < 0.001 vs. HTG/PVAT−; << *p* < 0.01 vs. HTG PVAT+/PPG−; >> *p* < 0.01 vs. HTG PVAT−/PPG−.

**Figure 2 biomolecules-11-00108-f002:**
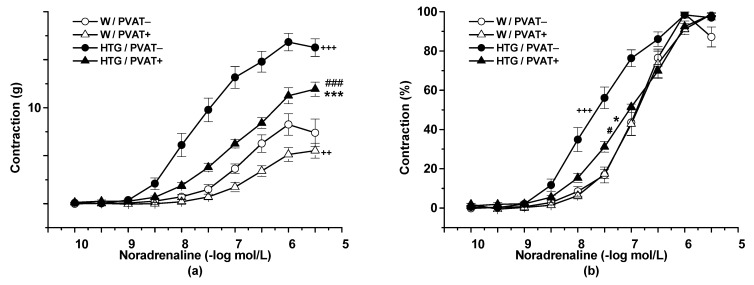
Noradrenaline-induced contractile response of the thoracic aorta of normotensive Wistar rats (W, n = 8) and hypertriglyceridemic rats (HTG, n = 9) with intact (PVAT+) or removed (PVAT−) perivascular adipose tissue expressed as the active wall tension (**a**) and as percentages of the maximal reached response induced by noradrenaline (**b**) before (PPG−) and after (PPG+) pretreatment with propargylglycine (**c**,**d**). The results are presented as the mean ± S.E.M, and differences between groups were analyzed by three-way ANOVA with the Bonferroni post hoc test on ranks. * *p* < 0.05 vs. W/PVAT+; *** *p* < 0.001 vs. W/PVAT+; +++ *p* < 0.001 vs. W/PVAT−; ++ *p* < 0.01 vs. W/PVAT−; # *p* < 0.001 vs. HTG/PVAT−; ### *p* < 0.001 vs. HTG/PVAT−; << *p* < 0.01 vs. HTG PVAT+/PPG−.

**Figure 3 biomolecules-11-00108-f003:**
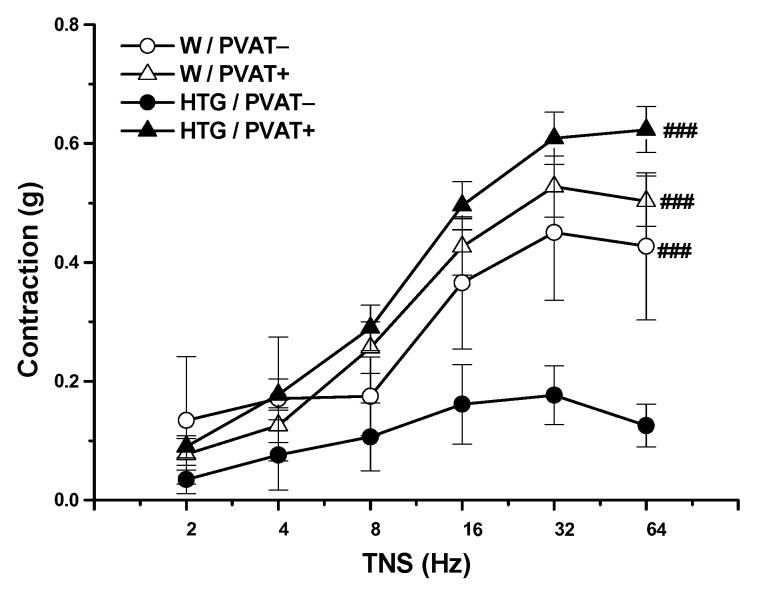
Contractile response of the mesenteric artery induced by endogenously released noradrenaline after transmural nerve stimulation (TNS) in normotensive Wistar rats (W, n = 8) and hypertriglyceridemic rats (HTG, n = 9) with intact (PVAT+) or removed (PVAT−) perivascular adipose tissue. The results are presented as the mean ± S.E.M, and differences between groups were analyzed by three-way ANOVA with the Bonferroni post hoc test on ranks. ### *p* < 0.001 vs. HTG/PVAT−.

**Figure 4 biomolecules-11-00108-f004:**
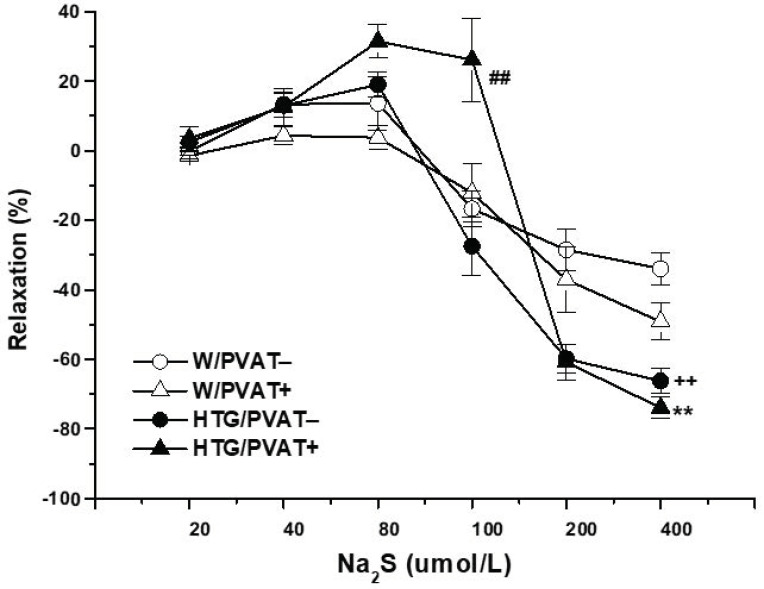
Dual vasoactive response of the thoracic aorta induced bysodium sulfide (Na_2_S),an exogenous donor of H_2_S in normotensive Wistar rats (W, n = 8) and hypertriglyceridemic rats (HTG, n = 9) with intact (PVAT+) or removed (PVAT−) perivascular adipose tissue. The results are presented as the mean ± S.E.M., and differences between groups were analyzed by three-way ANOVA with the Bonferroni post hoc test on ranks. ** *p* < 0.01 vs. W/PVAT+; ++ *p* < 0.01 vs. W/PVAT−; ## *p* < 0.01 vs. HTG/PVAT−.

**Figure 5 biomolecules-11-00108-f005:**
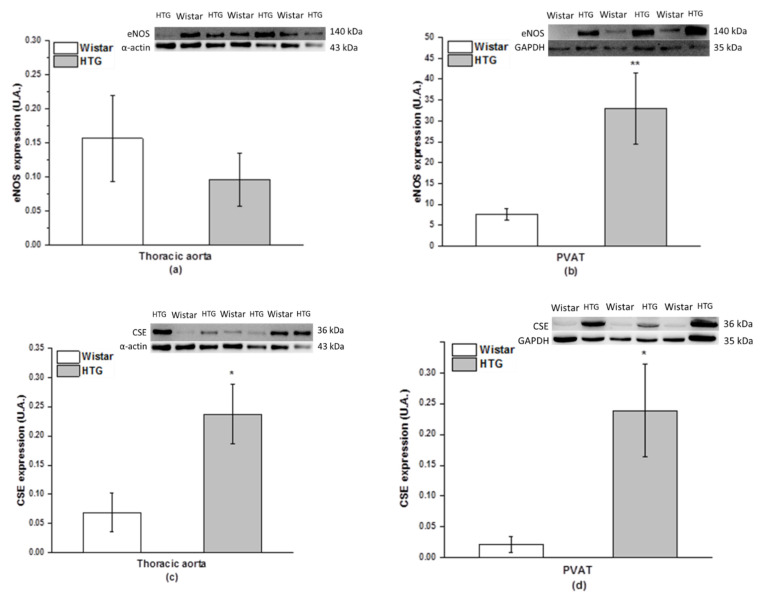
Protein expression of endothelial nitric oxide synthase (eNOS) in the aorta (**a**) and perivascular tissue (PVAT; **b**) and cystathionine gamma-lyase (CSE) in the aorta (**c**) and perivascular tissue (PVAT; (**d**) in Wistar rats (n = 7) and hypertriglyceridemic rats (HTG, n = 8). The results are presented as the mean ± S.E.M., and differences between groups were analyzed by one-way ANOVA with the Bonferroni post hoc test. * *p* < 0.05; ** *p* < 0.01 vs. HTG.

**Figure 6 biomolecules-11-00108-f006:**
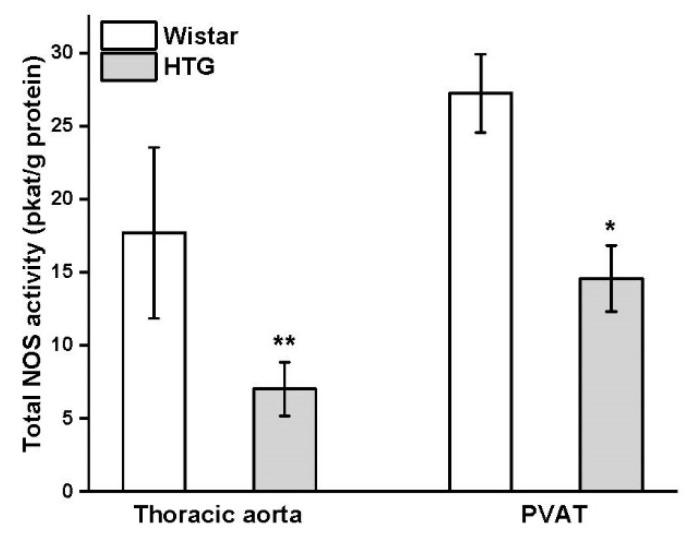
Total nitric oxide synthase (NOS) activity in the arterial wall of the thoracic aorta and perivascular tissue (PVAT) in Wistar rats (n = 7) and hypertriglyceridemic rats (HTG, n = 8). The results are presented as the mean ± S.E.M, and differences between groups were analyzed by one-way ANOVA with the Bonferroni post hoc test. * *p* < 0.05; ** *p* < 0.01 vs. HTG.

**Figure 7 biomolecules-11-00108-f007:**
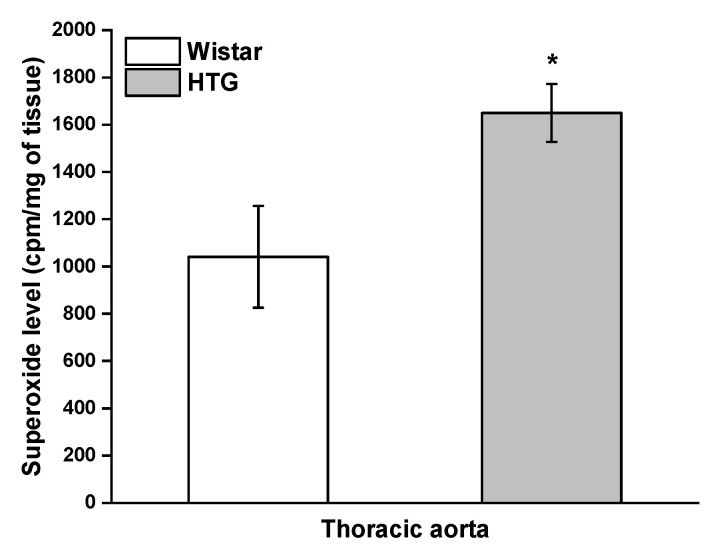
Superoxide level in the aortain Wistar rats (n = 7) and hypertriglyceridemic rats (HTG, n = 8). The results are presented as the mean ± S.E.M., and differences between groups were analyzed by one-way ANOVA with the Bonferroni post hoc test. * *p* < 0.05 vs. HTG.

**Figure 8 biomolecules-11-00108-f008:**
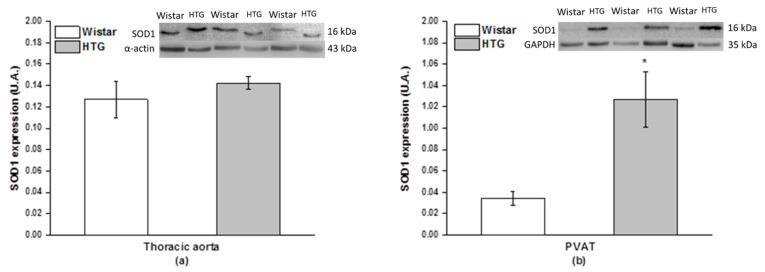
The protein expression of superoxide dismutase 1 (SOD1) in the aorta (**a**) and perivascular tissue (PVAT; **b**) and superoxide dismutase 2 (SOD2) in the aorta (**c**) and perivascular tissue (**d**) in Wistar rats (n = 7) and hypertriglyceridemic rats (HTG, n = 8). The results are presented as the mean ± S.E.M., and differences between groups were analyzed by one-way ANOVA with the Bonferroni post hoc test. * *p* < 0.05 vs. HTG.

**Table 1 biomolecules-11-00108-t001:** General characteristics of experimental animals.

Parameter	Wistar	HTG
SBP (mmHg)	118.48 ± 1.76	135.67 ± 2.14 *
BW (g)	425 ± 11	388 ± 10 *
HW (mg)	1.23 ± 0.29	1.06 ± 0.23
TL (mm)	39.48 ± 0.51	41.71 ± 0.38
HW/BW (mg/g)	2.93 ± 0.05	2.57 ± 0.09 *
HW/TL (mg/mm)	31.42 ± 0.66	25.47 ± 0.57 *
RFW (mg)	26.8 ± 0.28	45.0 ± 0.31 *
RFW/TL (mg/mm)	68.34 ± 6.98	108.16 ± 7.23 **
GLU (mmol/L)	4.8 ± 0.2	4.9 ± 0.2
OGTT AUC (mmol/L/2 h)	790 ± 58	907 ± 26 **
TAG (mmol/L)	1.57 ± 0.14	5.26 ± 0.45 ***
Chol (mg/dL)	1.76 ± 0.09	1.58 ± 0.05
HDL-C (mmol/L)	1.32 ± 0.07	0.57 ± 0.05 **
NEFA (mmol/L)	0.212 ± 0.012	0.410 ± 0.034 **
TAG in heart (μmol/g)	1.98 ± 0.23	2.42 ± 0.20
TAG in kidney (μmol/g)	3.45 ± 0.56	4.10 ± 0.44
MCP-1 (pg/mL)	170.0 ± 14.0	277.6 ± 30.8 *
TNFα (pg/mL)	1.67 ± 0.24	1.84 ± 0.15
IL-6 (pg/mL)	13.26 ± 1.78	15.49 ± 2.67
Leptin (ng/mL)	2.71 ± 0.13	3.56 ± 0.11 **
hsCRP (μg/mL)	220.9 ± 7.8	230.2 ± 16.7
ALT (U/L)	19.30 ± 2.67	33.12 ± 0.93 ***
Crea (μmol/L)	27.38 ± 0.68	31.56 ± 1.36 *
Urea (mmol/L)	5.90 ± 0.12	7.73 ± 0.19 ***

SBP, systolic blood pressure; BW, body weight; HW, heart weight; TL, tibia length; HW/BW, heart weight/body weight ratio; HW/TL, heart weight/tibia length ratio; RFW, retroperitoneal fat weight; RFW/TL, retroperitoneal fat weight/tibia length ratio; Chol, total cholesterol; HDL, high-density lipoprotein cholesterol; TAG, triacylglycerols; GLU, fasting glucose; OGTT AUC, the area under the glycemic curve during the oral glucose tolerance test; ALT, alanine-aminotransferase; Crea, creatinine; MCP-1, monocyte chemoattractant protein-1; TNFα, tumor necrosis factorα; NEFA, nonesterified fatty acids; IL-6, interleukin-6; hsCRP, high-sensitivity C-reactive protein. Values are shown as the mean ± S.E.M. * *p* < 0.05 vs. Wistar rats; ** *p* < 0.01 vs. Wistar rats; *** *p* < 0.001 vs. Wistar rats.

## Data Availability

All data arising from this study are contained within the article and any additional data sharing will be considered by the first author upon request.

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
