# Peer review of "The Vasoactive Role of Perivascular Adipose Tissue and the Sulfide Signaling Pathway in a Nonobese Model of Metabolic Syndrome"

_biomolecules, 2021, doi:10.3390/biom11010108_

Round 1

Reviewer 1 Report

The goal of this study was to examine and compare participation of PVAT and endogenously produced H2S in the contractile and relaxant responses of arteries isolated from normotensive Wistar rats and genetically fixed hypertriglyceridemic (HTG) rats. Vasoactive effects of expression of cystathione-ƴ-lyase, eNOS, total eNOS, and superoxide dismutase were also evaluated. In general, the authors did a good job on testing the two hypotheses they proposed in the paper. They found that endogenous H2S exerts compensatory vasoactivity in the thoracic PVAT; while endogenously produced H2S contributes to procontractile effects of PVAT in the mesenteric artery of HTG rats. They also explored the mechanism of how interaction between H2S pathway and PVAT leads to endothelial dysfunction in HTG rats. They presented comprehensive physiological data of HTG and Wistar rat. In addition, they conducted functional test of vasorelaxation and contraction with statistical rigor and appropriate control. The manuscript could be strengthened as described below.

Major comments:

  1. The authors used the hypotriglyceridemic (HTG) rats as a nonobese model. This rat shows typical phenotypes of hypertension, metabolic disorder such as glucose intolerance, and cardiac hypertrophy. However, the authors found that HTG rats showed hypotrophy (reduced HW/TL) of the myocardium in their experiment, which they suggest was a result of tissue injury. The author may need add to these data to supplementary results and they also need to exclude the influence of the abnormality of their models on results of functional studies.
  2. Add more methodological details of the anatomical segment of the aortae used for studies, as well as methods for removing PVAT.
  3. More detail is needed to explain the relaxation studies shown in Fig. 1. Because the contractile properties are different between Wistar and HTG rats (data presented in Fig. 2), it is important to know how the pre-contraction was achieved. What was the level of pre-contraction – was it determined based on percentage of maximal contraction? Was the contraction normalized to an assessment of overall viability (e.g. KCl contraction)? It might make more sense to describe the contraction first to set up for the relaxation studies.
  4. It would be very interesting to have a histological view of the PVAT comparing Wistar vs HTG rats. The observations that the HTG rats have increased retroperitoneal fat and chronic inflammation suggests that there could be pathological changes within the PVAT, which could be assessed.
  5. For western blot data, it is unclear how many samples are represented, and it would be helpful to include more than one sample to ensure consistency of the data. In Fig. 7d, the band intensity of SOD2 for HTG is visually similar to the band intensity of SOD2 protein for Wistar, which is not consistent with the corresponding quantification results. The images shown might not be representative of the entire group studied.
  6. The authors tested total NOS activity and protein expression of eNOS. However, they did not test cystathionine-γ-lyase (CSE) activity. Additional assays to measure CSE activity between experimental groups would make Fig. 6 more complete.

  1. The authors mention that their data contrast to previous publications that showed a reduced anti-contractile effect in models of metabolic dysfunction. Although they suggest one possibility that the nonobese HTG rats represent an early stage of prediabetes, confirmation of the pathology of the PVAT would be a good way to support this assertion (see point #2).

Minor comments:

  1. The authors might consider rewriting the abstract for better clarification of their experimental approach, key results, and final conclusion.
  2. It would be helpful to summarize all key findings in the beginning of the discussion section.
  3. Clarify which parameter is analyzed in Fig. 2b - percentage of what value?
  4. The authors can make their vascular reactivity data easier to read by incorporating symbols instead of long text inserts.
  5. In Table 1, the authors mislabeled the unit of HW/TL.
  6. In Fig. 4 legend, hydrogen sulfide should be sodium sulfide
  7. Add molecular weights to proteins detected on immunoblot.
  8. Define abbreviation of noradrenaline at first use.
  9. Quality of Fig. 5 is low, graphs can be made smaller to increase the representation of multiple samples/group for the immunoblot.
  10. Quality of panels in Fig. 7 are variable and different

Reviewer 2 Report

This study seeks to define the relationship between the vasoactive properties of the perivascular adipose tissue (PVAT) and hydrogen sulfide signaling in the PVAT and vessel wall. The authors use a hypertriglyceridemic rat as a model of metabolic syndrome and measure metabolic variables, assess vessel function in vitro, and quantify expression levels of key H2S, NO, and redox enzymes. Overall, the experiments are adequately controlled, and statistical analysis is appropriate. However, the study is very descriptive and there doesn’t appear to be any novel mechanistic insight, rather many if the results simply confirm what is already reported in the literature.

A major concern is that much of the interpretation rests on the use of a PPG as an inhibitor of CSE and Na2S as an H2S donor. This is problematic for a number of reasons. H2S is not measured, so how can any conclusion be made about levels of H2S in any of the experiments? Along similar lines, the concentrations of donor are likely to be nonphysiological, so what do these data mean? Even if inhibiting CSE decreases H2S production, interpretation of the effect is difficult - are the effects mediated by decreasing the release of H2S from PVAT or through indirect pathways? For example, does decreased H2S affect the release of some other paracrine mediator or adipokine? Is the effect of H2S mediated through immune cells that are in the PVAT? The uncertainty surrounding interpretation severely limits enthusiasm for this manuscript.  

Round 2

Reviewer 2 Report

The revisions made to the text are appropriate and the manuscript is somewhat improved by the more nuanced interpretation. However, the major concerns still remain because these concerns were predicated on the lack of experimental rigor and are therefore not fixable without additional experimental evidence.

The problem with the use of PPG and Ns2S still remain. There is no doubt that PPG is a useful tool that is commonly employed. The point is that a lot of the interpretation rests on this experiment. Without corroborating evidence, for example alternative inhibitors such as used in the Fang et al, 2009 paper, this approach lacks rigor. The same goes for the use of only one H2S donor.

Similarly, the critique that donors are non-physiological does not mean that the donors are useless, it just means that there is no way to use donors to recapitulate what is happening physiologically, or make any interpretation of what is happening physiologically. Just because H2S donors have an effect on any given target says nothing about how H2S functions physiologically because there is no way to know the actual concentration of H2S liberated by the donor.

Collectively, these limitations weaken the interpretation that H2S release from PVAT mediates the observed differences between W and HTG.